# Nano and Microemulsions for the Treatment of Depressive and Anxiety Disorders: An Efficient Approach to Improve Solubility, Brain Bioavailability and Therapeutic Efficacy

**DOI:** 10.3390/pharmaceutics14122825

**Published:** 2022-12-16

**Authors:** Patrícia C. Pires, Ana Cláudia Paiva-Santos, Francisco Veiga

**Affiliations:** 1Faculty of Pharmacy (FFUC-UC), University of Coimbra, Azinhaga de Santa Comba, 3000-548 Coimbra, Portugal; 2REQUIMTE/LAQV, Group of Pharmaceutical Technology, Faculty of Pharmacy, University of Coimbra, 3000-548 Coimbra, Portugal; 3Health Sciences Research Centre (CICS-UBI), University of Beira Interior, Av. Infante D. Henrique, 6200-506 Covilhã, Portugal

**Keywords:** anxiety, brain delivery, depression, microemulsion, nanoemulsion, solubility

## Abstract

Most drugs used for the treatment of depression, anxiety and related disorders have low absorption, high metabolism, low brain targeting and/or low water solubility, which can make it hard to formulate them at high strength and can also lead to decreased bioavailability. Incorporating these drugs into nanometric emulsions can solve these issues. Hence, the aim of the present review was to assess the potential of nano and micro emulsions for the delivery of antidepressant and anxiolytic drugs. The results from several studies showed that nanometric emulsions were able to increase drug strength up to 20,270-fold (compared to aqueous solubility). Moreover, in general, the formulations showed droplet size, polydispersity index, zeta potential, viscosity, osmolality, pH, in vitro drug release and ex vivo drug permeation as adequate for the intended effect and administration route. In vivo animal pharmacokinetic experiments showed that nanometric emulsions improved systemic drug bioavailability and/or brain targeting, and in vivo pharmacodynamic studies showed that they had antidepressant and/or anxiolytic effects, also being apparently safe. Hence, the current review provides proof of the potential of nano and microemulsions for improving solubilization and increasing the overall bioavailability of antidepressant and/or anxiolytic drugs, providing evidence of a possible useful strategy for future therapies.

## 1. Introduction

Depressive disorders are some of the most prevalent, impairing and costly illnesses, having recently been estimated to affect more than 246 million people worldwide [1,2,3]. Although they can be divided according to subtype and level of severity, these disorders are generally characterized by a depressed mood or general loss of pleasure or interest, usually accompanied by symptoms such as feelings of guilt, worthlessness or hopelessness; low self-esteem; indecisiveness or difficulty in concentrating or thinking; fatigue; psychomotor agitation or retardation; change in appetite; insomnia or hypersomnia; mood swings; and, in most severe cases, recurrent thoughts of death or suicidal ideation (Figure 1). The depressed person usually has a loss in work productivity and difficulty in managing social situations, with a decrease in general quality of life, and increased risk of suicide. Moreover, coexisting with other diseases can exacerbate their symptoms, being associated, for example, with a higher risk of severe complications in diabetic patients (amputation, blindness, dementia), and increasing the relative risk of developing heart disease by 200% [1,4,5]. Pharmacological treatment of depressive disorders can be quite variable, but selective serotonin reuptake inhibitors (fluoxetine, paroxetine, sertraline) are usually considered as first-line options. Other alternatives include serotonin and norepinephrine reuptake inhibitors (duloxetine, venlafaxine), tricyclic antidepressants (amitriptyline, imipramine, nortriptyline), tetracyclic antidepressants (mirtazapine), and atypical antidepressants (trazodone, nefazodone, agomelatine). Adjuvant hormonal and psychological treatments are also recommended in some cases, as well as lifestyle changes [1,4,6,7,8]. With so many available treatments to choose from, the choice should be made carefully, in a case-by-case trial-and-error approach, being individualized according to the patient’s characteristics (ethnicity, gender, age, socioeconomic status, comorbidities, etc.) and symptomatology [1,4].

Depression frequently coexists with other mental health disorders. There is a significantly increased risk of developing a comorbid depressive disorder when someone already has an anxiety disorder [1,5,9,10,11]. Anxiety disorders are also among the most common mental disorders, having been recently estimated to globally affect more than 265 million people. They are associated with substantial functional impairment, which leads to decreased work productivity and quality of life [2,9,12]. Aside from generalized anxiety disorder, there is a wide spectrum of related disorders (such as obsessive-compulsive, posttraumatic stress, panic and social anxiety disorders), but in general symptoms can include feeling exceptionally or chronically nervous, anxious or on edge; having excessive or uncontrollable fear and worry; sleep disturbances and hypervigilance; and constant avoidance of situations that relate to the previously mentioned symptoms (Figure 1) [9,11,13,14]. Patients with anxiety disorders also have a higher prevalence of other diseases, such as cardiovascular, respiratory and gastrointestinal conditions [9,12]. Treatment of anxiety and related disorders includes psychological and pharmacological options, and the choice again depends on patient related factors, such as severity of illness, prior treatment, comorbid disorders, patient preference and motivation, etc. [9]. The first-line pharmacological options are similar to those prescribed for depressive disorders: either selective serotonin reuptake inhibitors (escitalopram, fluoxetine, fluvoxamine, paroxetine, sertraline) or serotonin and norepinephrine reuptake inhibitors (duloxetine, venlafaxine), for being generally better tolerated and safer than other treatments [9,12]. Other options include noradrenergic and specific serotonergic antidepressants, tricyclic antidepressants, monoamine oxidase inhibitors, and reversible inhibitors of monoamine oxidase A [9,11]. Benzodiazepines can also be used, but as adjunctive short-term therapy, since they can cause dependency, sedation and cognitive impairment (especially with prolonged use) [9,12]. Some anticonvulsants and atypical antipsychotics have also demonstrated efficacy, but are generally recommended as second-line, third-line, or adjunctive therapies (due to side effects). Given the variety in treatment options, again it should be a case-by-case approach, taking into consideration efficacy versus safety, the specific characteristics of the anxiety disorder, comorbid conditions and treatment duration [9].

Yet, despite pharmacological treatment options for depressive and anxiety disorders being many, a great number of these drugs (including the grand majority of new drug candidates) have low water solubility, which can make it hard to formulate them at high strengths in liquid preparations [15]. This problem can be tackled by formulating these molecules into solid forms, with oral tablets being the most common option, but dose adjustment can sometimes be difficult, and inappropriate tablet splitting can lead to dose intake variation, which in turn can result in a reduction in treatment efficacy or exacerbation of adverse effects. Moreover, swallowing these formulations can be challenging, especially in the younger population (children and adolescents) or older individuals (particularly if having diseases linked to dysphagia, such as stroke, Parkinson’s, Alzheimer’s or cancer) [16,17]. Intravenous treatments require liquid solutions, but drug solubilization is usually achieved either by pH adjustments in the formulation, which if very low or very high could be potentially harmful, or using great amounts of organic cosolvents or surfactants, which are potentially toxic excipients, having been reported to cause hemotoxicity and hypersensitivity reactions (pruritus, erythema, rash or urticaria) [15]. Moreover, in these types of formulations, drugs are highly susceptible to metabolism, which can occur in all administration routes, but especially systemic ones, due to hepatic first-pass metabolism, being aggravated in oral administration, due to additional gastrointestinal degradation [18]. Additionally, in general, the treatment of central nervous systems disorders can be compromised by the very low permeability of the blood-brain barrier, which restricts the transport of most drug molecules, and this is especially relevant for the most common administration routes, in which the drug is transported to the brain from the bloodstream (oral and intravenous) [18,19].

Incorporating drugs into a nanosystem can be an effective strategy to tackle these issues. Nanosystems (also known as nanocarriers) are colloidal structures with a mean diameter of less than 500 nm. Among their many advantages, they allow: the enhancement of drug solubilization; metabolic and chemical degradation drug protection; the reduction of high plasma protein binding; increased permeability through biological membranes; and the promotion of brain bioavailability, which is especially useful for diseases with a brain etiology [19,20,21,22,23] (Figure 2). The many types of nanosystems can be divided into four main categories: polymeric nanocarriers, such as polymeric nanoparticles and micelles; lipid nanoparticles, such as solid lipid nanoparticles or nanostructured lipid carriers; liposomes and their derived counterparts, such as niosomes, ethosomes, transfersomes, cubosomes and phytosomes; and nanometric emulsions, such as nanoemulsions and microemulsions [18,24,25,26]. Yet, despite all mentioned advantages, most of these nanosystems have several drawbacks, such as a low encapsulation efficiency; reduced physical stability; requiring the use of organic solvents during preparation; requiring complex and time-consuming preparation methods; and having non-biocompatible components [27,28,29] (Figure 2).

However, nanometric emulsions can surpasses all of these drawbacks. Being colloidal liquid-in-liquid dispersions, they are usually made of a water phase, an oil, a surfactant, a cosurfactant and/or a cosolvent. They can be classified according to droplet size, between nanoemulsions (20,200 nm) or microemulsions (10–100 nm), although the size range can differ between authors. Regarding what concerns the differences in their characteristics, while microemulsions have thermodynamic stability, nanoemulsions have a relatively high kinetic stability, and both have higher surface area and free energy than macroemulsions, which makes them more physically stable in comparison. They can also be classified according to the nature of their internal and external phases, as: oil-in-water (*o*/*w*) or water-in-oil (*w*/*o*), if they are biphasic (most common); or oil-in-water-in-oil or water-in-oil-in-water, if they are triphasic (Figure 3) [18,30,31].

Nanometric emulsions are lipophilic, biocompatible, have a solubilizing effect and high encapsulation efficiency, which makes them promising options for encapsulating lipophilic drugs. Furthermore, formulas with the right composition (excipients and the proportion between them) can be very stable, and components such as surfactants and cosolvents allow them enhance drug permeation. In addition, unlike many other nanosystems, nano and microemulsions do not require the use of organic solvents during production, and aside from preparation methods that require high energy inputs (such as sonication, high pressure homogenization, extrusion through a small pore membrane), they can form spontaneously just by adding their components in specific proportions, which makes their preparation simpler, cheaper and less time-consuming than other nanocarriers, making them ideal for industrial application, where these are key factors (Figure 2) [18,30,31].

Therefore, this review aimed to collect detailed information and conduct a critical analysis on nano and microemulsions for the treatment of depressive and/or anxiety disorders, including formulation composition and characterization (droplet size, polydispersity index—PDI, zeta potential, viscosity, osmolality and pH), in vitro drug release, ex vivo drug permeation, in vivo pharmacokinetics, in vivo pharmacodynamics and/or safety studies (depending on the available data). The final goal was to summarize and analyze what has been conducted so far in this specific field, providing a source of information for future studies.

## 2. Nanometric Emulsions—Increasing Drug Solubility and Bioavailability through Several Administration Routes

Works developing nanometric emulsions containing antidepressant and anxiolytic drugs have been administered through several different routes, including intravenous, transdermal, oral and intranasal administration (Figure 4). All these administration routes have their advantages and disadvantages, which will be highlighted in the following Section 2.1, Section 2.2 and Section 2.3.

### 2.1. Nanometric Emulsions through Intravenous Administration—The Fastest Way to Achieve Systemic Drug Delivery

Intravenous administration involves injecting drugs directly into the bloodstream, which results in the fastest systemic drug delivery, being ideal for the treatment of acute and emergency situations. Moreover, since it bypasses any physical, chemical or biological barrier that might hinder drug absorption, it leads to the highest systemic bioavailability (theoretically 100%) among all delivery routes [32]. Nevertheless, this type of administration has its disadvantages, mostly due to the invasiveness of the injection, which can cause substantial discomfort or pain, and consequently decrease patient compliance, also having an associated risk of injury (and sometimes even infection) at the administration site. The intravenous route also requires trained personnel and, consequently, hospitalization, which is a major limitation [32,33]. In what concerns formulation characteristics, intravenous preparations should be sterile, isotonic (osmolality around 290 mOsmol/kg) and euhydric (physiological pH), in order to avoid local damage on vascular endothelium and circulating blood cells. These preparations should also have a low viscosity (up to 15 or 20 cP), since they should be easily drawn into a syringe and injected from it, and high viscosity intravenous formulations have been linked to blood viscosity increase, and consequently cardio or cerebrovascular adverse events [34].

Aripiprazole is an atypical third generation antipsychotic drug that acts mostly as an agonist for dopamine D2 and serotonin 5-HT1A receptors. It also has affinity for other receptors, such as dopamine D1, D3, D4 and D5 receptors, histaminergic H1, H2, H3 and H4 receptors, and adrenergic alpha-1 receptors, among others [35]. It has a reported low incidence of extrapyramidal side effects, and is used in the treatment of a wide variety of psychiatric disorders, such as schizophrenia, bipolar disorder, irritability associated with autism, Tourette’s syndrome and major depressive disorder [34,35]. Nevertheless, it has a very low water solubility (predicted to be 0.00777 mg/mL), and hence it is usually administered in solid forms (tablets) through the oral route [35]. However, given the generally faster therapeutic effect and higher bioavailability of the intravenous route (and the possibility of administration even in cases when swallowing is compromised), Samiun et al. [34] decided to formulate aripiprazole into an *o*/*w* nanoemulsion, using palm kernel oil esters, soybean lecithin (Lipoid S75), Tween^®^ 80, glycerol and deionized water (specific quantities summarized in Table 1). Aripiprazole was incorporated into the formulation at 0.10 *w*/*w*%, which is 128 times higher than the drug’s aqueous solubility. Lecithin was chosen as the primary emulsifier since it has been reported to aid drug transportation across the blood-brain barrier. Nanoemulsion preparation required high energy emulsification methods (high shear and high pressure homogenizers), and the obtained droplet size was 64.52 nm, viscosity 3.72 cP (within the established limits for intravenous preparations), osmolality 297 mOsm/kg (isotonic), and the pH was adjusted to 7.4 (neutral). The formulation was found to be reasonably stable under accelerated conditions (centrifugation) and after a 3-month storage at different temperatures. Nevertheless, while the authors consider the developed nanoemulsion to be a suitable carrier for the parenteral delivery of aripiprazole, and although the drug’s solubility in the preparation did in fact increase greatly when compared to its water solubility, no further studies were conducted. Hence, the question remains whether this nanoemulsion would have efficacy in delivering the drug to the brain or be therapeutically effective, and therefore future studies should be performed to address these issues, namely pharmacokinetic and/or pharmacodynamic in vivo experiments.

### 2.2. Transdermal Administration of Nanometric Emulsions—Overcoming the Skin Barrier

Transdermal administration delivers drugs across the skin’s layers to the blood circulatory system. It can be preferred over the parenteral route for being non-invasive, thus circumventing its associated issues, such as needle phobia [33,36]. When compared to the oral route, it has the advantages of avoiding hepatic first-pass metabolism and gastrointestinal degradation, which can increase drug bioavailability, and not needing repeated dosing, which can increase patient compliance, also being suitable for patients for whom the oral route is not eligible (situations such as unconsciousness or vomiting) [21,32,33,36]. By providing sustained drug plasma levels, transdermal delivery is especially suitable for drugs that need relatively constant plasma levels and prolonged duration of the therapeutic effect [21,32]. It is also associated with more uniform pharmacokinetic drug profiles, with fewer peaks, thus minimizing the risk of toxic side effects [33]. Nevertheless, in order to be feasible candidates for delivery by transdermal administration, drugs should have certain characteristics, such as being highly potent, small in size (<500 Da), and having a log *p*-value between 1 and 3 (lipophilic) [21,32,33]. Moreover, transdermal delivery is associated with overall poor drug permeation through the skin barrier, which can consequently lead to low bioavailability [36]. Hence, some methods can be used to temporarily and reversibly modify the skin barrier: physical methods, such as iontophoresis, electroporation and ultrasound; or chemical methods, such as the use of excipients with absorption enhancing capability (e.g., fatty acids, surfactants, terpenes and solvents). Nevertheless, these methods should be used with caution, since they could cause toxicity and skin irritation [21,32].

The decrease in the endogenous hormone allopregnanolone during pregnancy has been linked to post-partum depression. Brexanolone is a neurosteroid that acts by mimicking this hormone, making it the first and only treatment specifically against this type of depression [37,38]. Brexanolone is believed to have a barbiturate-like activity, by being a positive allosteric modulator of both synaptic and extrasynaptic GABA type A receptors, hence enhancing GABA activity by increasing the opening of GABA type A receptor calcium channels, and for keeping them open for a longer period of time [38]. Nevertheless, it has very low water solubility (predicted to be 0.00136 mg/mL), and is approved for use as an intravenous infusion only, which requires hospitalization [37,38]. Hence, Bhattaccharjee et al. [37] decided to formulate brexanolone into a microemulsion, for transdermal administration, aiming for sustained drug delivery. Two types of microemulsions were developed: one *o*/*w* and one *w*/*o*, with composition being chosen according to a drug solubility screening in individual excipients. Both microemulsions were made of oleic acid (Super Refined^TM^ Oleic Acid NF), Labrasol^®^, Plurol^®^ Oleique CC 497, Transcutol^®^ P and deionized water, with the quantities of each component differing between them (values in Table 2). The drug strength was 10 mg/mL for the *o*/*w* microemulsion and 19 mg/mL for the *w*/*o* microemulsion, which is 7353 and 13,971 times higher than brexanolone’s water solubility, respectively. The obtained droplet sizes were 129 ± 0.208 nm, with a PDI of 0.123 ± 0.007, for the *o*/*w* microemulsion, and 136 ± 0.291 nm, with a PDI of 0.149 ± 0.009, for the *w*/*o* microemulsion. The microemulsion preparation included a step of mixing through vortexing, and both formulations were considered to be stable under accelerated conditions (centrifugation). The transdermal delivery of these microemulsions was assessed in an ex vivo permeation study conducted in microporated skin (dermatomed human skin pieces, with laser ablation pre-treatment), and resulted in a significantly higher drug delivery when compared to a propylene glycol drug solution. Moreover, despite the *w*/*o* microemulsion having been able to solubilize more drug, the *o*/*w* microemulsion had a significantly higher transdermal delivery. This might have been due to the *o*/*w* microemulsion having a greater number of surfactants and cosolvents in its composition (when compared to the *w*/*o* microemulsion), which have been described to have permeation enhancing effects.

### 2.3. Oral Delivery of Nanometric Emulsions—Overcoming the Problems Related to the Most Common Route

Whenever possible to use, non-invasive administration methods are usually the best option for chronic therapy. Within them, the oral route is the most common, being conventionally chosen to deliver the great majority of small molecular weight drugs [32,33]. The ease of self-administration, painlessness and cost-effectiveness associated with this route all lead to high patient compliance [32,39]. Moreover, drugs can have access to a large surface area available for absorption to the systemic circulation, with the possibility for sustained and controlled delivery [39]. Nevertheless, orally delivered drugs need to deal with multiple levels of barriers. Prior to absorption, the harsh environment of the gastrointestinal tract can lead to chemical and enzymatic drug degradation, and after absorption the first-pass hepatic metabolism can also significantly reduce drug bioavailability. Moreover, the oral route is not suitable for emergency situations, due to drug absorption being generally slow, and for situations in which swallowing is not possible [32,40].

Duloxetine is a potent dual inhibitor of serotonin and norepinephrine reuptake, and less potent inhibitor of dopamine reuptake [41]. It is used to treat fibromyalgia, neuropathic pain, generalized anxiety disorder and depression, but has a very low water solubility (predicted to be 0.00296 mg/mL) [41,42]. In order to overcome this issue, Sindhu et al. [42] tried incorporating it into an *o*/*w* microemulsion for oral administration, for the treatment of depression. Excipients were selected according to drug solubility, and among them were Capmul^®^ MCM, Tween^®^ 80, Transcutol^®^ P and water (detailed quantities in Table 3). The achieved drug strength was 60 mg/mL, which is 20,270 times higher than duloxetine’s water solubility. The microemulsion was obtained through spontaneous emulsification, and the measured droplet size was 35.40 ± 3.11 nm (Appendix A), PDI 0.17, zeta potential −25.8 mV, viscosity 0.205 cP, and pH 5.6 ± 0.5 (within the acceptable range for oral delivery). The preparation was also considered to be stable under physical stress conditions (heating-cooling cycles, freeze-thaw cycles and centrifugation). In vitro drug release studies (dialysis bag method) showed that the developed microemulsion had a faster and overall higher release than a duloxetine suspension (same drug strength, with 2% *w*/*v* of sodium carboxymethyl cellulose). In the ex vivo permeation study (rat duodenum) the permeability of the duloxetine microemulsion was also significantly higher (1.5 times) when compared to the suspension (Appendix A). In vivo pharmacokinetics in rats (oral administration) showed that the duloxetine systemic bioavailability obtained with the microemulsion was 1.8 times higher than the obtained with the drug suspension, with a maximum drug concentration (C_max_) that was also more than twice as high (Appendix A). These results could be due to the fact that the microemulsion was able to solubilize the drug (whereas the suspension was not), but also due to the absorption enhancing capability of the nanometric emulsion itself (small droplet size, large surface area) and its components (namely the surfactant and the cosolvent). In depression, due to the serotonin deficiency, there is a direct impact on motor dysfunction. Hence, depressed animals will have reduced motor and behavioral activity, and in vivo pharmacodynamic tests, such as mobility test, forced swimming test and tail suspension test are adequate models to evaluate the efficacy of the developed formulations containing the antidepressant drug. Hence, in this study mobility improvement after administration was assessed in depression-induced rats. Results showed that the developed oral microemulsion was also more effective than the oral drug suspension or vehicle, which can be concluded as being directly related to a more effective brain drug transport, leading to higher serotonin levels and, consequently, antidepressant action.

Vitamin E’s antioxidant properties have been linked to many beneficial health effects, including in cardiovascular diseases, cancer and neuroprotection. Low levels of this vitamin have also been associated with memory, cognitive and emotional disorders [43,45]. Although vitamin E’s exact mechanism of action is still unknown, in general it has proven to prevent free radical reactions with cell membranes, avoiding lipid peroxidation [45]. Nevertheless, because it is highly hydrophobic, it has very low solubility in aqueous fluids (predicted water solubility 0.00000704 mg/mL), including those that exist in the gastrointestinal tract [43,45]. Hence, Wilhelm et al. [43] decided to develop a vitamin E microemulsion for oral administration, for the treatment of depression and anxiety disorders. The microemulsion was made by spontaneous emulsification, and aside from vitamin E (1.00% *w*/*v*) it had Span^®^ 80, Tween^®^ 80 and distilled water in its composition (detailed quantities in Table 3). The droplet size was 306.3 ± 3.1 nm, and zeta potential was −29.4 ± 2.5 mV. In vivo experiments consisted of a once per day administration to mice, through the intragastric route, for an 8-day period (subchronic treatment). On the 8th day, behavioral tests showed that the vitamin E microemulsion had antidepressant-like and anxiolytic-like effects that were more evident than any other comparative formulation (vehicles and free vitamin E). Moreover, the microemulsion reduced lipid peroxidation levels more than any other given treatment, which suggested that these therapeutic effects were at least partially related to its antioxidant effect. In what concerns toxicity, there were no signs of the developed formulation causing oxidative damage or altering hepatic functions.

AJS (code name) is a novel antidepressant drug based on the structure of cinnamamide (Tasly Holding Group Company, Tianjin, China). It acts on serotonin and the noradrenaline receptors, but has a low oral bioavailability due to having poor water solubility (0.0049 mg/mL). In order to tackle this issue, Wu et al. [44] developed a self-microemulsifying drug delivery system (SMEDDS), containing castor oil, Labrasol^®^, Cremophor EL^®^ and Transcutol^®^ HP (excipients selected according to drug solubility, specific quantities in Table 3), with the drug at 3.40% *w*/*w*, which is 6939 times higher than its aqueous solubility. SMEDDS are oil, surfactant and cosurfactant (and/or cosolvent) mixtures that when in contact with the gastrointestinal tract’s fluids will form a microemulsion spontaneously. Since they do not have water in their composition, they tend to have higher stability than nanometric emulsions (formed prior to administration). The developed SMEDDS’ droplet size (after dilution) was found to be 26.08 ± 1.68 nm, with a PDI value of 0.264 ± 0.01, a zeta potential of −2.76 ± 0.27 mV, and a viscosity of approximately 264 cP. As for stability, no drug precipitation or phase separation was observed after a 3-month storage at 25 °C. In vivo pharmacokinetics in rats showed that the oral administration of the developed SMEDDS, when compared to a hydroxypropyl methylcellulose-based solid dispersion and β-cyclodextrin inclusion complexes of drug (also administered orally), led to a systemic bioavailability that was 3.4- and 35.9-fold greater (respectively). Moreover, the blood C_max_ that was obtained with the SMEDDS was also significantly greater when compared with the other two formulations (2.2 times higher than the dispersion, and 35.5 times higher than the inclusion complexes). These results were probably due to the larger surface area obtained from the droplet formation upon dilution in the gastrointestinal tract of the oral SMEDDS, and having surfactants and a cosolvent in its composition, which are known permeation enhancers. Moreover, the authors claim that the drug also underwent intestinal lymphatic transport (known to happen for lipidic formulations), which made it possible for at least part of the drug to reach the systemic circulation without having to pass through the liver (first-pass metabolism).

### 2.4. Intranasal Nanometric Emulsions—A Direct Route to the Brain

Intranasal administration is promising for the treatment of affections with a brain etiology due to allowing (at least part of) the drug to reach the brain directly by neuronal transport. This also makes it possible for drugs to simultaneously (at least partially) avoid the blood-brain barrier, the harsh environment of the gastrointestinal tract and the hepatic first-pass metabolism. Therefore, it can not only increase brain drug bioavailability and minimize systemic adverse events, but also generate a short onset of action, which is a must in emergency situations. Moreover, the intranasal route is non-invasive and the formulations can be easily administrated by the patients themselves or a caregiver, hence not requiring hospitalization. Additionally, it is a good alternative to the oral route for patients with symptoms such as vomiting, increased salivation, or inability to swallow. Nasal liquid or semisolid preparations should have non-irritant components, a pH between 5.0 and 6.5 (similar to the nasal mucosa’s), and be isotonic to slightly hypertonic. The limitations associated with this administration route include requiring a low administration volume (150–200 μL for humans, therefore requiring relatively potent drugs), the possibility of the formulation’s residence time in the nasal cavity being short (which could be tackled by increasing the formulation’s viscosity or adding a mucoadhesive polymer), and the presence of degrading enzymes and efflux transporters in the nasal cavity [18,20,46].

Buspirone’s anxiolytic activity is thought to be related to its action as a serotonin 5-HT1A receptor agonist, acting as a full agonist of presynaptic 5-HT1A receptors, and as a partial agonist of postsynaptic 5-HT1A receptors. It also has a weaker affinity for other receptors, acting as an agonist for serotonin 5-HT2 receptors, antagonist for dopamine D2, D3 and D4 receptors, and partial agonist for adrenergic alpha-1 receptors [47]. Unlike other anxiolytic drugs, it does not exhibit anticonvulsant, sedative, hypnotic or muscle-relaxant properties, therefore being a selective anxiolytic agent. Being a polar molecule, it has poor permeability, but its water solubility is also not very high (predicted to be 0.588 mg/mL). Furthermore, this drug has low oral bioavailability due to poor absorption and extensive first-pass metabolism [47,48]. For these reasons, Bshara et al. [48] developed *o*/*w* microemulsions, to be administered intranasally for the treatment of anxiety. Excipients were selected based on their reported ability to improve brain targeting, increase polar drugs’ absorption, and their capacity for buspirone (hydrochloride) solubilization. Three formulas were selected from preliminary tests, all having isopropyl myristate, Tween^®^ 80, propylene glycol and water in their composition (quantities shown in Table 4). One of them additionally had chitosan aspartate as a mucoadhesive, and a third formula also had chitosan aspartate plus hydroxypropyl-β-cyclodextrin (absorption enhancer). The drug was kept at 1% *w*/*w*, which is 17-fold higher than its water solubility. Measured droplet sizes were between 30 and 40 nm (Appendix A), PDI values between 0.13 and 0.15, and zeta potential values between −3 and +6 mV. The addition of chitosan aspartate caused a significant increase in microemulsion viscosity (286.9 ± 5.3 cP to 343 ± 4.7 cP), but the addition of hydroxypropyl-β-cyclodextrin did not increase it further in a substantial way. The presence of chitosan aspartate also caused a 1.3-fold increase in the microemulsion’s mucoadhesive strength, and the presence of hydroxypropyl-β-cyclodextrin caused an additional 1.7-fold increase. The developed formulations were stable up to 6 months (storage under different temperatures) and under accelerated conditions (centrifugation). Ex vivo permeation studies (sheep nasal tissues) showed that the cumulative drug permeation (after 6 h) of the drug solution (29.59%) was less than half than the obtained with the non-mucoadhesive microemulsion (65.15%), which in turn was lower than the achieved with the mucoadhesive microemulsion (75.5%) and mucoadhesive microemulsion with hydroxypropyl-β-cyclodextrins (100%). This suggested that the microemulsion’s composition (namely surfactant and cosolvent), the mucoadhesive polymer and the cyclodextrins all caused an enhancement in drug permeation. As for in vivo pharmacokinetics (in rats), all the intranasal microemulsions had higher brain C_max_ and area under the “drug concentration vs. time” curve (AUC) values than the intravenous or intranasal drug solutions, with the mucoadhesive microemulsion with cyclodextrins being better than the mucoadhesive microemulsion (without cyclodextrins), which in turn was better than the non-mucoadhesive microemulsion at making the drug reach the brain (Appendix A). Finally, in what concerns formulation safety (rat nasal mucosa histopathological examinations), after daily administration for 7 days, no severe signs of necrosis, sloughing of epithelial cells or hemorrhage were detected (Appendix A).

Clobazam is a benzodiazepine derivative used for epilepsy, schizophrenia and anxiety treatment. Along with its active metabolite, norclobazam, it acts as a partial agonist to GABA-A receptors, binding to them allosterically, to their α and γ2-subunit interface. This will increase the frequency of the chloride channel opening, and also membrane permeability to chloride ions, leading to a hyper polarization and stabilization of the neuronal membrane, hence enhancing the post-synaptic inhibitory effect of GABA [51]. Due to its lipophilic nature and high protein binding, it has high oral bioavailability and a long half-life, which can be good for therapeutic effect, but has also been linked to various systemic adverse events (gastrointestinal disturbances, muscular spasms, irregular heartbeats), drug tolerance and dependence [49]. Moreover, it has low water solubility (predicted to be 0.164 mg/mL) [51]. Hence, in order to have a selective and fast delivery of clobazam to the brain, while reducing associated systemic adverse events, Florence et al. [49] developed clobazam *o*/*w* microemulsions, for intranasal delivery. The non-mucoadhesive microemulsion contained Capmul^®^ MCM, Acconan^®^ CC6, Tween^®^ 20, and distilled water (specific quantities in Table 4). The mucoadhesive microemulsion had the same composition, except for the addition of the mucoadhesive polymer Carbopol^®^ 940P. Drug strength was 3 mg/mL for both microemulsions, which is more than 18 times higher than clobazam’s water solubility. As for formulation characteristics, the droplet sizes were 16.47 ± 5.4 nm and 19.79 ± 6.2 nm, PDI 0.168 and 0.181, zeta potential −8.45 ± 5.05 mV and –15.2 ± 3.46 mV, and viscosity 7.73 ± 0.43 cP and 25.8 ± 0.71 cP, for non-mucoadhesive and mucoadhesive microemulsions, respectively, with a pH between 5 and 6. The addition of Carbopol to the formula led to a lowering of the zeta potential (more negative) and increase in viscosity, as was expected of an anionic and viscosifying polymer. Both formulas appeared to be stable under accelerated test conditions (centrifugation and freeze-thaw cycles). Ex vivo permeation studies (sheep nasal mucosa), in which the microemulsions were compared to a drug solution (containing propylene glycol, polyethylene glycol, ethanol and Tween^®^ 20), showed that the microemulsions had an increased permeation. These studies also showed that, despite being more viscous, the mucoadhesive microemulsion had the highest drug permeation, which is explained by the authors as being due to the presence of Carbopol, which leads to an opening of the tight junctions that exist in the nasal mucosa. The in vivo pharmacokinetic studies (mice) showed that brain AUC and C_max_ values for the intranasal microemulsions were higher than those obtained for intranasal and intravenous solutions, with the mucoadhesive microemulsion having the highest values. Moreover, the lower blood AUC and C_max_ values obtained with the intranasal microemulsions, when compared with the intravenous solution, make them potentially safer in what concerns systemic side effects (Appendix A). The mucoadhesive microemulsion also had the highest brain/blood drug ratios at all time points (when compared to the other formulations), which shows enhanced brain uptake (Appendix A). An evidently higher and selective accumulation in the brain was shown for the intranasal route, and most significantly for the developed nanosystems (Appendix A).

Paroxetine is a selective serotonin reuptake inhibitor used to treat mood disorders such as panic, obsessive compulsive, major depressive and generalized anxiety disorder. It also shows some affinity to muscarinic cholinergic receptors, and weak affinity to adrenergic alpha-1, alpha-2 and β receptors, dopamine D1 and D2 receptors, and histamine H1 receptors. It has low oral bioavailability due to undergoing extensive first pass metabolism, and is also poorly soluble in aqueous fluids (predicted water solubility 0.00853 mg/mL) [52]. Therefore, Pandey et al. [50] decided to develop an *o*/*w* nanoemulsion, for intranasal administration, to treat depression. Excipients were selected according to highest paroxetine solubility, and included Capmul^®^ MCM, Solutol HS 15, propylene glycol and distilled water (quantities specified in Table 4). The drug was incorporated at 1.6% *w*/*w*, which is 1876 times higher than its aqueous solubility. The nanoemulsion was obtained using the spontaneous emulsification method, with a droplet size of 58.47 ± 3.02 nm, PDI of 0.339 ± 0.007, zeta potential of −33 mV, and viscosity of 40.85 ± 6.40 cP. The formulation appeared to be stable under physical stress conditions (heating-cooling cycles, freeze-thaw cycles and centrifugation). The ex vivo permeation study (porcine nasal mucosa) showed that the developed nanoemulsion led to a 2.57-fold permeation enhancement in comparison to a paroxetine suspension.

The therapeutic efficacy of the developed formulation was assessed in pharmacodynamic studies (chronic depression induced rats), as they can provide useful information on the formulation’s potential in reducing the symptoms of depression. Results showed that the intranasal administration of the nanoemulsion led to a significant improvement in behavioral activity (increased swimming time, climbing time and locomotor activity, and reduction in immobility time), performing better than an orally administered drug suspension. Moreover, biochemical estimation tests showed that the intranasal paroxetine nanoemulsion led to a reduction in generated reactive oxygen species, and also increased the level of glutathione. Furthermore, a histopathological examination of brain tissues showed that the nanoemulsion decreased neuronal degeneration, erosion and damage in rats (with chemically-induced depression), having a protective role.

### 2.5. Final Remarks

Nanometric emulsions can be delivered through a variety of administration routes, though their characteristics (viscosity, osmolality, pH) and composition (especially surfactant and cosolvent amounts) should be adapted, in order to obtain optimum efficacy and safety. Moreover, not all drugs are suited for all administration routes, since some routes require potent drugs (transdermal and intranasal), and drugs meant to be administered in emergency situations, in which the therapeutic effect should be reached as fast as possible, cannot be administered through routes that are linked to slow absorption (oral and transdermal). Hence, when choosing a formulation composition and deciding on its characteristics target profile, the nanometric emulsion development process should always account for disease intended to treat, administration route and drug attributes.

#### 2.5.1. Antidepressant and Anxiolytic Drugs—Chosen Molecules for Nanometric Emulsion Incorporation

In what concerns studied drugs (and other substances with potential pharmacological effect), most were already approved for the treatment of one or more depressive, anxiety or related disorders, being dopamine, serotonin and/or norepinephrine receptor agonists or reuptake inhibitors, GABA-A receptors partial agonists or neurosteroids. Two exceptions were one novel antidepressant drug molecule (action on serotonin and noradrenaline receptors, not further specified) and one nutraceutical (antioxidant properties). The nano and microemulsions that were developed to incorporate these molecules were able to increase their solubility up to 20,270-fold (compared to their aqueous solubility). A summary of these molecule’s names and structure is depicted in Table 5.

#### 2.5.2. Nanometric Emulsion Types, Preparation Methods and Formulation Characterization Parameters

Nanometric emulsions are generally advantageous when compared to other nanosystems in what concerns simplicity of preparation, since they can be formed spontaneously with the right excipients in the right proportions. Nevertheless, in the studies included in this review, not all formulations were prepared by spontaneous emulsification, with one being prepared by high energy emulsification methods (high shear homogenizer and high pressure homogenizer) [34], and another by mixing through vortexing [37] (although this last one is still a quite simple procedure). While nanometric emulsions with good characteristics could be achieved by using more complex methods than a simple mixture of their components, a simpler method equals lower costs and time consumption in production, which could be essential in a pharmaceutical industry context. As for nanometric emulsion subtypes, this review reports them as named by the authors, although looking at the measured droplet sizes one might question this classification in one or two cases, if having as reference the values given in the introduction section: 10–100 nm for microemulsions and 20–200 nm for nanoemulsions. Nevertheless, other factors should be considered carefully when classifying these types of formulations, such as, for example, the decrease in droplet size that happens with increasing dilution in the case of microemulsions (data which was not provided by any of the included works).

Parameters such as droplet size, PDI, zeta potential, viscosity, osmolality and pH should always be measured and reported, since they could greatly influence nanometric emulsions’ efficacy and/or safety. All articles included in this review reported droplet size, but only six out of eight reported the PDI, which is an essential parameter to determine the homogeneity of the formulation, and can directly influence its stability, also affecting drug absorption and distribution. Zeta potential was also frequently reported (six out of eight), but it is especially important in cases when it is expected to not be neutral (neutral excipients tend to lead to neutral values), since high absolute values have been known to lead to a higher formulation stability. Moreover, high values can be especially relevant in specific routes of administration, such as in intranasal delivery, in which a high positive zeta potential can result in interactions (electrostatic adsorption) between the nanometric emulsions’ droplets and the negatively charged sialic acid residues of the nasal mucosa, which can help retain the preparation at the administration site. As for viscosity, it was reported in six out of eight studies: for the intravenous nanoemulsion, being low, which is recommended in order to be easily drawn into or injected from a syringe, and not lead to blood viscosity increase (and consequent adverse events); and for all intranasal nanometric emulsions, which is important since it can also affect the easiness of administration (depending on the administration device), with the formulations preferably having a viscosity that is high enough to increase their retention time in the nasal cavity (consequently increasing the time available for drug absorption to occur), but not so high that it limits drug diffusion from the preparation itself (unless a sustained drug release is intended). One oral and one transdermal microemulsion did not measure this parameter, which is relevant in the case of the oral route since it could limit drug diffusion from the preparation, with a high viscosity only being good if a sustained drug release is intended (but not for immediate release), and being an essential parameter in the case of the transdermal route, due to a sustained release being generally required. As for osmolality and pH, they were only reported in one out of eight and three out of eight studies, respectively, which represents a low frequency, especially since these parameters could directly correlate with formulation safety, being particularly important when no safety studies were performed. As for formulation stability, it is important to evaluate the good condition of the developed preparation throughout time, or under specific physical stress conditions, being especially relevant when considering a transition to the pharmaceutical industry and potentially reaching the market. Most authors assessed for formulation stability, evaluating physical stability through visual observation, or by measuring the formulation’s characterization parameters again (droplet size, PDI, etc.), after storage at different temperatures for a specific amount of time, or under accelerated conditions (heating-cooling or freeze-thaw cycles, and/or centrifugation).

#### 2.5.3. In Vitro Drug Release and Ex Vivo Drug Permeation

In what concerns drug release and permeation, only one out of eight articles conducted in vitro drug release studies (oral microemulsion) [42], but five out of eight articles conducted ex vivo drug permeation studies (transdermal, one oral and all intranasal microemulsions) [37,42,48,49,50]. Although in vitro drug release studies could be simpler to perform, and be useful for understanding the extent and speed of drug release from the developed formulations, ex vivo permeation studies, albeit more complex, could provide information more closely related to the in vivo situation, since they are performed in excised tissues (and not synthetic membranes). Moreover, although ex vivo permeation studies do not directly study drug release, studying formulations with a similar composition but, for example, having different viscosities, can originate a slower or more limited drug permeation, which could be inferred as being related to a more sustained drug release. Ideally, if possible, one should do both, in order to obtain drug release and drug permeation data. If conducted before in vivo pharmacokinetic or pharmacodynamic studies, these studies could help understand, deepen the knowledge or predict the outcomes of animal experiments. Moreover, if more than one formulation is being considered, they could also help choose between them, leading to a reduction in animal use.

#### 2.5.4. In Vivo Pharmacokinetics and Pharmacodynamics and Complemenetary Biochemical Assays

Albeit being generally optimistic models, in vivo animal studies, whether evaluating drug distribution (pharmacokinetic) or therapeutic efficacy (pharmacodynamic), are essential to assess the true potential of a given drug and/or formulation, being the closest to predicting drug/formulation performance in humans without actually performing clinical trials. Neither the intravenous aripiprazole nanoemulsion [34] nor the transdermal brexanolone microemulsions [37] had this type of evaluation, which leaves a knowledge gap. The oral duloxetine microemulsion [42] was evaluated for both pharmacokinetics and pharmacodynamics, but in the pharmacokinetic study it was only compared to an oral drug suspension, with no intravenous control (systemic bioavailability 100%, could have provided additional useful information regarding drug absorption). Moreover, the drug was only quantified in the rats’ plasma, not in the brain, and although systemic bioavailability can already be a good indication of potential efficacy, the direct evaluation of brain drug distribution lets us know whether the drug has reached its intended therapeutic site, and if so to what extent. Nevertheless, the pharmacodynamic study showed that the formulation had efficacy in improving mobility in depression induced rats, which is an indication that the drug did in fact reach the brain. In pharmacokinetic studies in rats, the oral AJS SMEDDS [44] was also only compared to other orally administered preparations (hydroxypropyl methylcellulose-based solid dispersion and β-cyclodextrin inclusion complexes), again lacking a comparative intravenous group and determination of brain drug levels (only blood). Yet, in this case, no pharmacodynamic study was conducted, so the efficacy that the developed formulation had in making the drug reach the brain is more difficult to assess. Although no pharmacokinetic studies were performed, in vivo pharmacodynamics in mice was conducted for the oral vitamin E microemulsion [43], and it had antidepressant-like and anxiolytic-like effects (more than the controls—vehicles and free vitamin E). As for the studies regarding intranasally administered formulations, two of them were evaluated for in vivo pharmacokinetics, and both the buspirone [48] and the clobazam [49] microemulsions had intranasal and intravenous control groups, also measuring both blood and brain drug levels, which made it possible to conclude that the developed preparations had good brain targeting and performed better than intranasal or intravenous drug solutions. The intranasally administered paroxetine nanoemulsion [50] was evaluated for its in vivo pharmacodynamics in chronic depression induced rats, and, although it was only compared to an orally administered drug suspension (neither intranasal nor intravenous control groups), it led to an improvement in behavioral activity, performing better than the oral suspension. Hence, in the articles that reported in vivo studies, in general the developed nano or microemulsions led to an improved systemic drug bioavailability and/or brain targeting (when compared to the controls, independently of whether these controls could be considered the most adequate). This is probably mainly due to the excipients that are part of these formulation’s composition, which increase drug permeation (surfactants, cosolvents, cyclodextrins) and/or increase formulation retention at the administration site (mucoadhesives, specifically in the case of intranasal administration). Moreover, studies that included in vivo pharmacodynamics showed that the developed nanometric emulsions had antidepressant and/or anxiolytic effects (also being more effective than the controls). Since the true effectiveness of a given formulation can only be assessed in animal models (before studies in humans), the articles that did not perform them have a grand limitation, with the true potential of the produced nanosystems being left undetermined. It is highly recommended that all studies perform these assays, especially behavioral studies, since they are the only in which therapeutic-like effectiveness can be evaluated.

The information provided by pharmacokinetic and/or pharmacodynamic in vivo studies could also be complemented by results from specific biochemical assays. The oral vitamin E microemulsion [43] appeared to have its antidepressant-like and anxiolytic-like effects (at least partially) linked to its antioxidant effect, since it led to a significant reduction in lipid peroxidation levels (more than controls). Additionally, the intranasal paroxetine nanoemulsion [50] also seemed to have its antidepressant effects connected to antioxidant properties, since it led to a reduction in generated reactive oxygen species and increased glutathione levels.

#### 2.5.5. Safety Studies

Only two out of eight articles specifically studied formulation safety. The oral vitamin E microemulsion [43] showed no signs of causing oxidative damage or altering hepatic functions in mice (biochemical determination after in vivo pharmacodynamic study), and the intranasal buspirone microemulsions [48] did no damage to the rats’ nasal mucosa (histopathological examinations after daily administration for 7 days). Formulation safety can sometimes be overlooked, with most studies focusing on therapeutic efficacy only, but a therapeutically effective formulation that causes a great number of side effects, especially if severe, might not have a favorable risk/benefit ratio. Hence, this parameter should be assessed. Yet, although not performing safety studies, many of the articles included in this review chose excipients having safety aspects into account, either by: choosing excipients that are classified as “generally recognized as safe” (GRAS); searching for excipient safety data prior to selection; minimizing the amount of potentially toxic excipients included in the formulation (namely surfactants); or adjusting formulation pH and/or osmolality (and sometimes even viscosity, in the case of intravenous formulations) according to the requirements for the intended administration route. Despite not being as reassuring as performing actual safety studies, taking these aspects into consideration could lead to the development of potentially safer formulations.

## 3. Conclusions

The development of nanometric emulsions to encapsulate antidepressant and anxiolytic drugs has proven to be effective in increasing both drug strength and delivery, especially for lipophilic molecules. This happens not only due to small droplet size and the possibility of encapsulation of said molecules, but also due to the use of excipients with solubilizing capacity and permeation enhancing properties, such as surfactants, cosolvents and cyclodextrins. Furthermore, formulation characterization is not complete without determining and reporting droplet size, PDI, zeta potential, viscosity, osmolality and pH, which are all factors that could influence their in vivo performance and/or safety. Formulation stability studies are also recommended in order to know the time during which a selected formula will keep its properties. In vitro drug release, ex vivo drug permeation and specific biochemical estimations are not as indispensable, but might provide useful information that could help explain, deepen the knowledge or predict the outcomes of in vivo studies. On the other hand, in vivo animal pharmacokinetic and/or pharmacodynamic experiments are essential in order to assess the full potential of a developed formulation, and without them that assessment is left incomplete. Safety studies should also be more frequently performed, since even if a certain formulation is therapeutically effective, it is not promising unless it has a reasonably favorable efficacy/safety ratio. Hence, although the number of studies that have been performed so far is still small, which presents a limitation for drawing generalized conclusions, overall, nano and microemulsions have shown to be promising strategies to improve the solubilization and increase the bioavailability of antidepressant and/or anxiolytic drugs, being potential strategies to replace current therapies. More experimental studies should be conducted in the future, including clinical trials, in order to address these formulations’ true medical applicability.

## Figures and Tables

**Figure 1 pharmaceutics-14-02825-f001:**
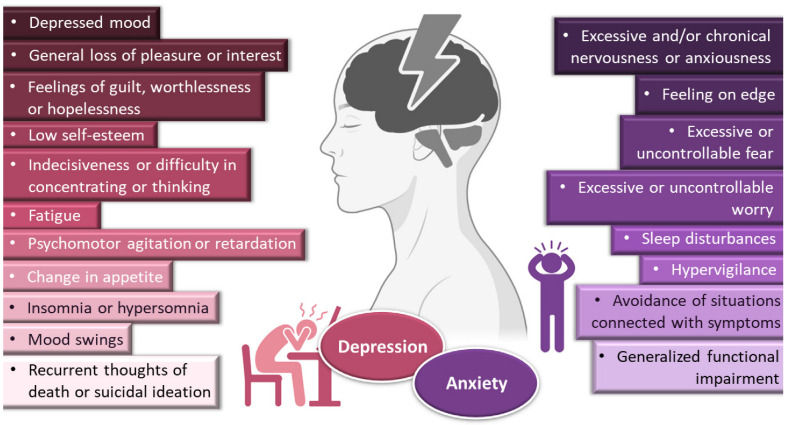
General symptoms of depression and anxiety disorders. Drawn with BioRender (no copyright required).

**Figure 2 pharmaceutics-14-02825-f002:**
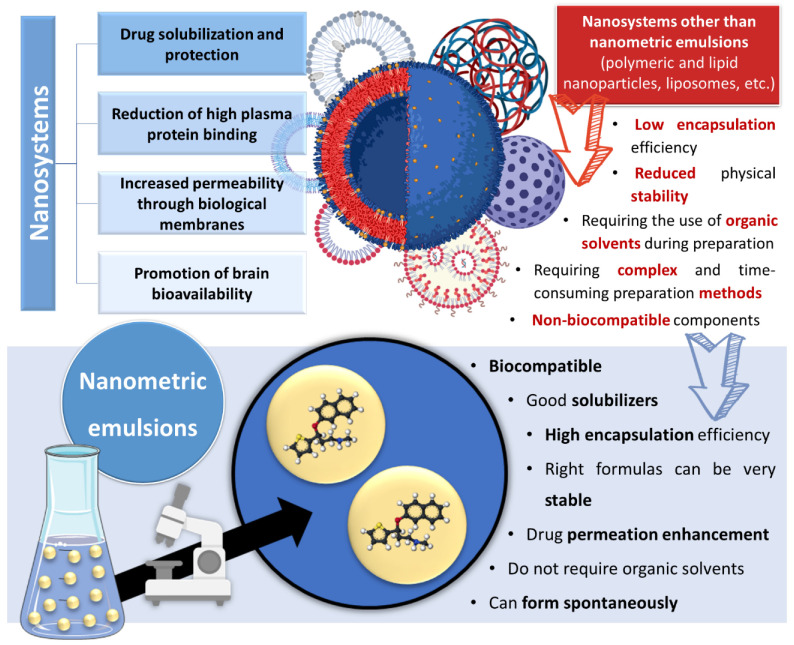
Advantages and disadvantages of drug encapsulation into nanosystems, with a focus on the superiority of nanometric emulsions. Drawn with BioRender (no copyright required).

**Figure 3 pharmaceutics-14-02825-f003:**
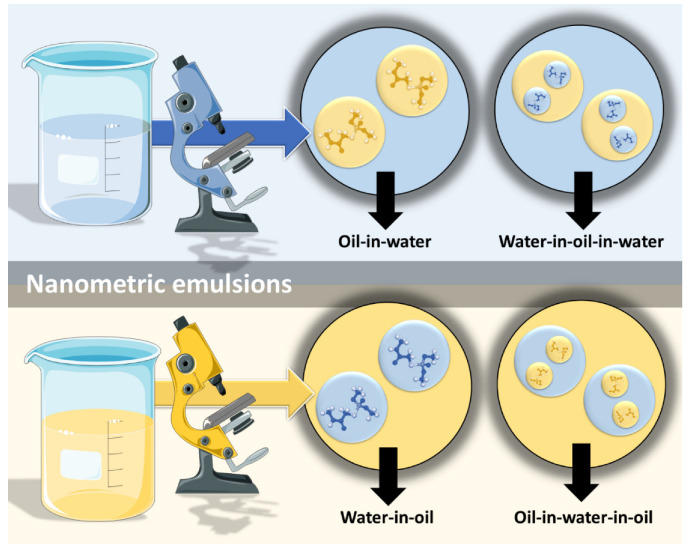
Types of nanometric emulsions, according to the nature of their internal and external phases. Drawn with BioRender (no copyright required).

**Figure 4 pharmaceutics-14-02825-f004:**
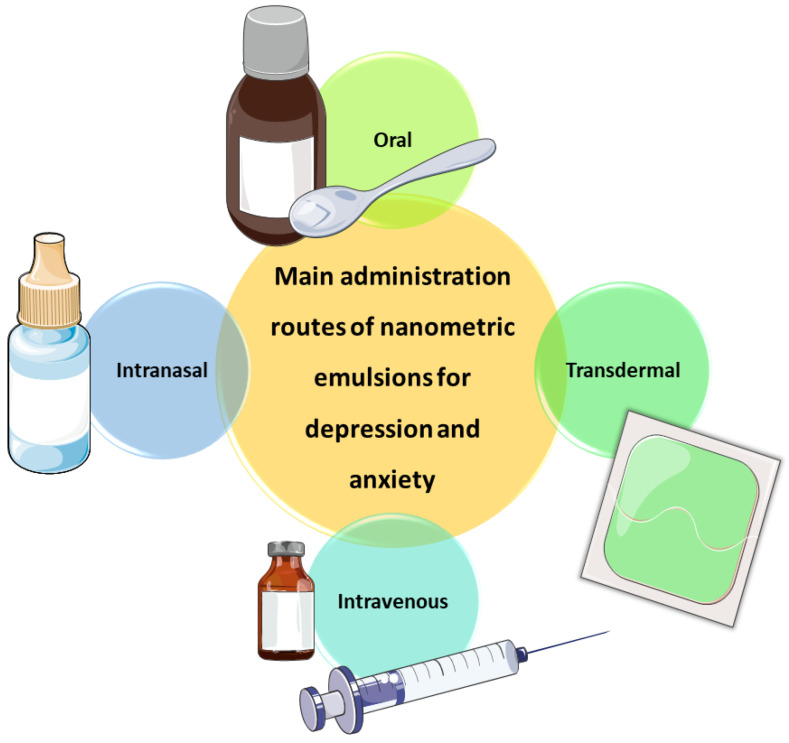
Administration routes through which nanometric emulsions containing antidepressant and anxiolytic drugs have been administered. Drawn with BioRender (no copyright required).

**Table 1 pharmaceutics-14-02825-t001:** Detailed composition of the intravenous nanoemulsion developed by the works included in this review. Excipient and drug quantities and units are shown as reported in the respective articles (*w*/*w*%). For the excipients, usually the brand name was used (when available).

Composition		Intravenous Aripiprazole *o*/*w* Nanoemulsion [34]
Oil	Palm kernel oil esters	3.00% *w*/*w*
Hydrophobic surfactant	Lipoid S75 ^1^	2.00% *w*/*w*
Hydrophilic surfactant	Tween^®^ 80 ^2^	1.00% *w*/*w*
Cosolvent	Glycerol	2.25% *w*/*w*
Water	-	91.75% *w*/*w*
Drug	Aripiprazole	0.10% *w*/*w*

^1^ soybean lecithin; ^2^ polysorbate 80—polyoxyethylene 20 sorbitan monooleate.

**Table 2 pharmaceutics-14-02825-t002:** Detailed composition of the transdermal microemulsions developed by the works included in this review. Excipient and drug quantities and units are shown as reported in the respective articles (*w*/*w*% for excipients, and mg/mL for drugs). For the excipients, usually the brand name was used (when available).

Composition		Transdermal Brexanolone Microemulsions [37]	
		*o*/*w*	*w*/*o*
Oil	Oleic acid	7.00% *w*/*w*	10.00% *w*/*w*
Hydrophobic surfactant	Plurol^®^ Oleique CC 497 ^1^	7.14% *w*/*w*	11.43% *w*/*w*
Hydrophilic surfactant	Labrasol^® 2^	42.86% *w*/*w*	68.57% *w*/*w*
Cosolvent	Transcutol^®^ P ^3^	17.20% *w*/*w*	4.00% *w*/*w*
Water	-	25.80% *w*/*w*	6.00% *w*/*w*
Drug	Brexanolone	10 mg/mL	19 mg/mL

^1^ polyglyceryl-3 dioleate; ^2^ polyethylene glycol-8 caprylic/capric glycerides; ^3^ diethylene glycol monoethyl ether.

**Table 3 pharmaceutics-14-02825-t003:** Detailed composition of the oral microemulsions and SMEDDS developed by the works included in this review. Excipient and drug quantities and units are shown as reported in the respective articles (*w*/*v* or *w*/*w*% for excipients, and *w*/*w*% or mg/mL for drugs). For the excipients, usually the brand name was used (when available).

Composition		Oral Duloxetine *o*/*w* Microemulsion [42] *	Oral Vitamin E Microemulsion [43]	Oral AJS SMEDDS [44]
Oil	Capmul^®^ MCM ^1^	10.00%	-	-
	Vitamin E	-	1.00% *w*/*v*	-
	Castor oil ^2^	-	-	24.49% *w*/*w*
Hydrophobic surfactant	Span^®^ 80 ^3^	-	ND	-
Hydrophilic surfactant	Tween^®^ 80 ^4^	32.00%	ND	-
	Labrasol^® 5^	-	-	28.57% *w*/*w*
	Cremophor EL^® 6^	-	-	40.82% *w*/*w*
Cosolvent	Transcutol^®^ P or HP ^7^	8.00%	-	2.72% *w*/*w*
Water	-	50.00%	ND	-
Drug	Duloxetine	60 mg/mL	-	-
	AJS ^8^	-	-	3.40% *w*/*w*

ND—concentration not described for final formulation; SMEDDS—self-microemulsifying drug delivery system; * percentual units not specified; ^1^ medium chain mono- and diglycerides; ^2^ triglyceride of fatty acids, mostly ricinoleic acid; ^3^ sorbitan monooleate; ^4^ polysorbate 80—polyoxyethylene 20 sorbitan monooleate; ^5^ polyethylene glycol-8 caprylic/capric glycerides; ^6^ polyoxyl 35 castor oil; ^7^ diethylene glycol monoethyl ether; ^8^ code name for novel drug.

**Table 4 pharmaceutics-14-02825-t004:** Detailed composition of the intranasal nano and microemulsions developed by the works included in this review. Excipient and drug quantities and units are shown as reported in the respective articles (*v*/*v*, *w*/*v* or *w*/*w*% for excipients, and *w*/*w*% or mg/mL for drugs). For the excipients, usually the brand name was used (when available).

Composition		Intranasal Buspirone *o*/*w* Microemulsions [48]			Intranasal Clobazam *o*/*w* Microemulsions [49]		Intranasal Paroxetine *o*/*w* Nanoemulsion [50]
		Non-Mucoadhesive	Mucoadhesive	Mucoadhesive with Cyclodextrins	Non-Mucoadhesive	Mucoadhesive	
Oil	Isopropyl myristate	5.00% *w*/*w*	5.00% *w*/*w*	5.00% *w*/*w*	-	-	-
	Capmul^®^ MCM ^1^	-	-	-	5.00% *v*/*v*	5.00% *v*/*v*	8.00% *
Hydrophobic surfactant	Acconan^®^ CC6 ^2^	-	-	-	22.50% *v*/*v*	22.50% *v*/*v*	-
Hydrophilic surfactant	Tween^®^ 80 ^3^	30.00% *w*/*w*	30.00% *w*/*w*	30.00% *w*/*w*	-	-	-
	Tween^®^ 20 ^4^	-	-	-	7.50% *v*/*v*	7.50% *v*/*v*	-
	Solutol HS 15 ^5^	-	-	-	-	-	21.00% *
Cosolvent	Propylene glycol	15.00% *w*/*w*	15.00% *w*/*w*	15.00% *w*/*w*	-	-	21.00% *
Mucoadhesive and/or viscosifying agents	Chitosan aspartate	-	0.10% *w*/*w*	0.10% *w*/*w*	-	-	-
	Carbopol^®^ 940P ^6^	-	-	-	-	0.50% *w*/*v*	-
Other components	Hydroxypropyl-β-cyclodextrin	-	-	1.00% *w*/*w*	-	-	-
Water	-	50.00% *w*/*w*	50.00% *w*/*w*	50.00% *w*/*w*	65.00% *v*/*v*	65.00% *v*/*v*	50.00% *
Drug	Buspirone (hydrochloride)	1.00% *w*/*w*	1.00% *w*/*w*	1.00% *w*/*w*	-	-	-
	Clobazam	-	-	-	3 mg/mL	3 mg/mL	-
	Paroxetine	-	-	-	-	-	1.6% *w*/*w*

* percentual units not specified; ^1^ medium chain mono- and diglycerides; ^2^ mixture of polyoxyethylene-6-caprylic and capric glycerides; ^3^ polysorbate 80—polyoxyethylene 20 sorbitan monooleate; ^4^ polysorbate 20—polyoxyethylene 20 sorbitan monolaurate; ^5^ macrogol 15 hydroxystearate; ^6^ carbomer 940.

**Table 5 pharmaceutics-14-02825-t005:** Names and chemical structures of the antidepressant and anxiolytic drug molecules’ incorporated in the nanometric emulsions included in this review.

Drug Molecule Name	Chemical Structure	Formulation	Administration Route	Ref.
Aripiprazole	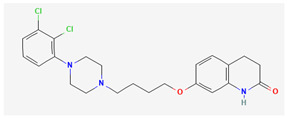	*o*/*w* nanoemulsion	Intravenous	[34,53]
Brexanolone	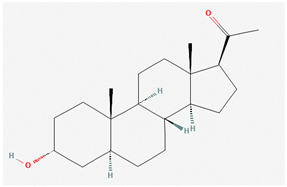	*o*/*w* and *w*/*o* microemulsions	Transdermal	[37,54]
Duloxetine	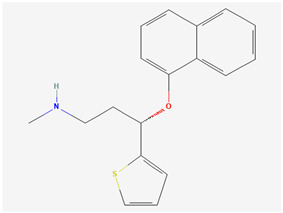	*o*/*w* microemulsion	Oral	[42,55]
Vitamin E	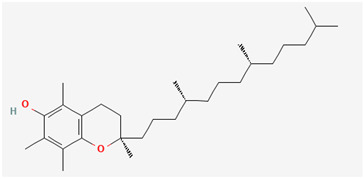	microemulsion	Oral	[43,56]
AJS	Not disclosed (confidential information)	SMEDDS	Oral	[44]
Buspirone	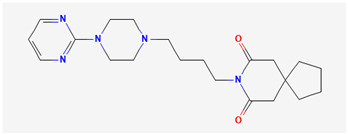	*o*/*w* microemulsions	Intranasal	[48,57]
Clobazam	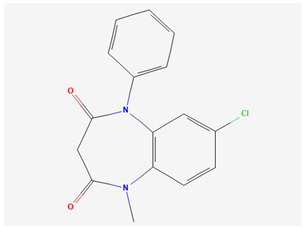	*o*/*w* microemulsions	Intranasal	[49,58]
Paroxetine	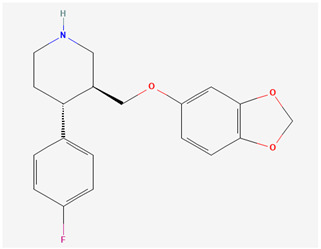	*o*/*w* nanoemulsion	Intranasal	[50,59]

## Data Availability

Not applicable.

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
