# Peer review of "Nano and Microemulsions for the Treatment of Depressive and Anxiety Disorders: An Efficient Approach to Improve Solubility, Brain Bioavailability and Therapeutic Efficacy"

_pharmaceutics, 2022, doi:10.3390/pharmaceutics14122825_

Round 1
Reviewer 1 Report
The manuscript describes nanoemulsions in the therapy of depression. Nanoemulsions are promising agents for the therapy of disorders, including neuropsychiatric disorders. Therefore, the topic is relevant and a lot of research is being done in this field.
However, this work has some disadvantages.
I recommend to revise the English, because already in the abstract, the sentences do not sound very good.
The abstract is more like an introduction. It should describe what the review is about: what each part of the review is about and how the review can be useful for readers.
The review lacks a lot of data on the effectiveness of nanoemulsions. It is mentioned only once in brief on page 11. There is a lot of data about the routes of administration, their pros and cons, nanocontainer systems and pharmacokinetics. Meanwhile, none of this matters if the drug is not effective. There is clearly not enough data on this in animal models or in humans in the manuscript. I recommend to insert this information into the text.
Figure 1 has no name. What does the lightning in figure 1 symbolize?
Figure 2 is also unclear. What are "nanosystems" and "other types of nanosystems"? The authors need to write more details on what they are talking about.
It is unclear how the literature data was collected for this review? Was the review systematic and were all articles on the topic reviewed? A search in Pubmed for articles on the query "nаnоemulsion depression" and "microemulsion depression" found 10 and 18, correspondingly. So it is quite possible to look through all the articles on this topic, maybe the authors did, but it is not mentioned anywhere.
The Final remarks are very long and not structured. It should be shortened at least by half. Perhaps some sentences should be moved to the parts above, such as lines 621-628, 643-649 and others. Final remarkst should also be divided into subheadings. Or at the discretion of the authors they can make a table with conclusions about the data to make the manuscript more understandable.
I think it's better not to insert pictures from other studies. At least not in the main text of the articles, if the authors want to keep them, move them to the supplement.
Author Response
We thank the reviewer for their constructive criticism and expert opinion. The manuscript has been altered and improved according to all the reviewers suggestions (changes marked by coloring in blue), and a point-by-point answer is given below.
“The abstract is more like an introduction. It should describe what the review is about: what each part of the review is about and how the review can be useful for readers.” – We thank the reviewer for their comment. The abstract has now been completed and restructured according to the suggestions.
“The review lacks a lot of data on the effectiveness of nanoemulsions. It is mentioned only once in brief on page 11. There is a lot of data about the routes of administration, their pros and cons, nanocontainer systems and pharmacokinetics. Meanwhile, none of this matters if the drug is not effective. There is clearly not enough data on this in animal models or in humans in the manuscript. I recommend to insert this information into the text.” – We thank the reviewer for their comment. The effectiveness of nanoemulsions is illustrated throughout the manuscript, as each study is described and critically analyzed. Hence, the effectiveness of the nanoemulsions in delivering each antidepressant and/or anxiolytic drug molecule to the brain, and having therapeutic-like efficacy, is mentioned from lines 317 to 326, 350 to 357, 372 to 384, 435 to 441, 485 to 495, and 512 to 518. Furthermore, the overall efficacy of these formulations is also mentioned, from lines 624 to 667.
“Figure 1 has no name. What does the lightning in figure 1 symbolize?” – We thank the reviewer for their comment. The caption was included in the submitted version of the manuscript, must have been an editing error. Figure 1 caption has now been added.
“Figure 2 is also unclear. What are "nanosystems" and "other types of nanosystems"? The authors need to write more details on what they are talking about.” – We thank the reviewer for their comment. “Nanosystems” refers to all categories, in general, and “other types of nanosystems" refers to nanosystems other than nanometric emulsions. We understand it might not have been completely clear, and hence have made adaptations to the figure accordingly.
“The Final remarks are very long and not structured. It should be shortened at least by half. Perhaps some sentences should be moved to the parts above, such as lines 621-628, 643-649 and others. Final remarkst should also be divided into subheadings. Or at the discretion of the authors they can make a table with conclusions about the data to make the manuscript more understandable.” – We thank the reviewer for their comment. The “Final remarks” section has now been divided into sub-sections, now having an organized and coherent structure, in order to facilitate reader comprehension.
“I think it's better not to insert pictures from other studies. At least not in the main text of the articles, if the authors want to keep them, move them to the supplement.” – We thank the reviewer for their comment, and have moved these images to the Supplementary Materials, as suggested.
Reviewer 2 Report
The manuscript highlights the use of nano or microemulsions as more advantageous systems for solubilizing and deliverying drugs used for anxiety and depressive disorders, which use to present low aqueous solubility. Considering the small number of studies cited and in each table (table 1, 2 and 3), the reader can be confused about the real advantages of these systems. As the manuscript was submitted to an Special Issue of Innovative Formulations of Poorly Soluble Drugs, it is worth including other formulations and the outcomes of these strategies. Since the focus is the antidepressive drugs, the sections could be arranged by drugs and not by administration routes, showing the comparison of nano/microemulsions and other systems. Figure 1 and Figure 3 seem unecessary. Although interesting and well written, the manuscript is quite extensive and in case the focus in nano/microemulsions is maintained, it should be shortened.
Author Response
We thank the reviewer for their constructive criticism and expert opinion. The manuscript has been altered and improved according to all the reviewers suggestions (changes marked by coloring in blue), and a point-by-point answer is given below.
“Considering the small number of studies cited and in each table (table 1, 2 and 3), the reader can be confused about the real advantages of these systems.” – We thank the reviewer for their comment. It is a fact that there aren’t many studies regarding nanometric emulsions encapsulating antidepressant and anxiolytic drugs. Nevertheless, overall the already existing literature on the subject supports that these nanosystems can be quite promising for the treatment of these pathologies, since not only do they allow drug strength in formulation to be increased up to thousands-fold, but they also improve brain drug delivery, when compared to drug solutions or suspensions. Hence, the existing studies on the subject suggest strong promise for these formulations. Nevertheless, the fact that there is still a small number of studies is a limitation, and more studies should be done in the future, including clinical trials. Hence, a paragraph stating this has been added to the manuscript, from lines 712 to 718.
“As the manuscript was submitted to an Special Issue of Innovative Formulations of Poorly Soluble Drugs, it is worth including other formulations and the outcomes of these strategies.” – We thank the reviewer for their comment. The focus of the present review is nano and microemulsions specifically, their compositions, preparation methods, characterization parameters and efficacy regarding in vivo evaluation. The focus is, therefore, the pharmaceutical technology aspects of this type of nanosystem, for the delivery of antidepressant and anxiolytic drugs. It is an approach that is still lacking in the scientific literature, hence being innovative, and we believe it could be of great interest to scientists in the field, also fitting right within the scope of this journal. Therefore, including individual studies regarding other types of nanosystems (such as nanoparticles, liposomes, etc.) would change that completely. Yet, regarding the comparison of nanometric emulsions with other types of nanosystems, this information has been added/completed from lines 111 to 125, 130 to 137, and 144 to 154, also being present in Figure 2.
“Since the focus is the antidepressive drugs, the sections could be arranged by drugs and not by administration routes, showing the comparison of nano/microemulsions and other systems.” – We thank the reviewer for their comment. The current structure was thought having in mind the fact that formulation characteristics should be adapted according to the administration route they are intended for. Since the current review is focused on the pharmaceutical technology aspects of the developed formulations, we thought of this structure as best suited, especially concerning the scope of the journal Pharmaceutics. Furthermore, by grouping the studies by administration route, it is easier to see what the tendency has been in what concerns the favored administration route for nanometric emulsion containing antidepressant and/or anxiolytic drugs - intranasal administration. Hence, the organization by drug molecule could make sense in other types of articles, but here we feel it is more relevant by grouping according to administration route.
“Figure 1 and Figure 3 seem unecessary.” – We thank the reviewer for their comment. The figures were made in order to help any reader better understand the content of the paper, in this case the most important information on the addressed pathologies, and the included administration routes. Therefore, we feel that it is beneficial for these figures to exist, and that their existence is an advantage for better comprehension, since they summarize and systematize important information. Furthermore, these types of figures are quite common in the reviews published by this journal, and hence their existence fits right within the journal’s requirements and scope.
“Although interesting and well written, the manuscript is quite extensive and in case the focus in nano/microemulsions is maintained, it should be shortened.” - We thank the reviewer for their comment. As mentioned by the reviewer in the first comment, “Considering the small number of studies”, we sought to thoroughly report and critically analyze all the included articles, so that in the future scientists in the field can resort to our article in order to find important pharmaceutical technology and efficacy information regarding these formulations. Hence, we feel that this moderately extensive analysis is a benefit, and also provides innovation regarding other reviews on similar subjects, since any researcher that reads our paper will be able to immediately assess the potential of each formulation, and reproduce it if needed.
Reviewer 3 Report
The manuscript has dedicated to an interesting topic in the drug delivery field, nano-systems for delivery of anti-depressant and anti-anxiety drugs. Overall, the manuscript is useful and well prepared. However, the authors should be improved the manuscript by addressing the following issues.
1. The authors should spend more time for mentioning about mechanism action of anti-depressant and anti-anxiety drugs, issues that all drugs and delivery systems meet for targeted to central nervous system due to brain blood barrier, and particularly, mechanism for releasing drugs at target sites of microemulsions and nanoemulsions.
2. The difference between microemulsion and nanoemulsion in term of physicochemical properties and preparation method should be complemented with more references.
3. I suggest the author separating the part 2.1 into two sections, because the intravenous administration and transdermal administration are considerably difference, besides, in the intravenous injection, the system is directly distributed to blood, while transdermal route required the drugs or nanocarrier must pass throughout many barriers (cells, tissues).
4. Advantages and disadvantages of microemulsion and nanoemulsion in comparison with other nanocarriers, for example polymeric NPs, inorganic NPs, lipid based NPs, etc, need to be clarified.
5. In each section (administration routes), the authors should discuss about essential requirements for successfully delivering therapeutics, including size of NPs, characteristic of surface, stability, etc.
6. Fig 1 missed name and explanation.
7. For make a better understanding, the authors can provide a table that lists name and chemical structure of anti-depressant and anti-anxiety drugs.
Author Response
We thank the reviewer for their constructive criticism and expert opinion. The manuscript has been fully altered and improved according to all the reviewers suggestions (changes marked by coloring in blue), and a point-by-point answer is given below.
“The authors should spend more time for mentioning about mechanism action of anti-depressant and anti-anxiety drugs, issues that all drugs and delivery systems meet for targeted to central nervous system due to brain blood barrier, and particularly, mechanism for releasing drugs at target sites of microemulsions and nanoemulsions.” – We thank the reviewer for their comment. This information have now been added from lines 106 to 110, from lines 192 to 194, 252 to 255, 299 to 300, 340 to 342, 403 to 407, 457 to 460, and 498 to 500.
“The difference between microemulsion and nanoemulsion in term of physicochemical properties and preparation method should be complemented with more references.” – We thank the reviewer for their comment. References were in fact lacking in this paragraph, but this has now been corrected.
“I suggest the author separating the part 2.1 into two sections, because the intravenous administration and transdermal administration are considerably difference, besides, in the intravenous injection, the system is directly distributed to blood, while transdermal route required the drugs or nanocarrier must pass throughout many barriers (cells, tissues).” - We thank the reviewer for their comment. Section 2.1 has been divided according to the suggestions.
“Advantages and disadvantages of microemulsion and nanoemulsion in comparison with other nanocarriers, for example polymeric NPs, inorganic NPs, lipid based NPs, etc, need to be clarified.” - We thank the reviewer for their comment. This information is present and has been completed in Figure 2 and from lines 111 to 125, 130 to 140, and 144 to 154.
“Fig 1 missed name and explanation.” – We thank the reviewer for their comment. The caption was included in the submitted version of the manuscript, must have been an editing error. Figure 1 caption has now been added.
“For make a better understanding, the authors can provide a table that lists name and chemical structure of anti-depressant and anti-anxiety drugs.” - We thank the reviewer for their comment. An additional Table has been inserted, according to the suggestions (Table 5, pages 15 to 17), including studied drug name and structure, and also the corresponding formulation type and administration route.
Round 2
Reviewer 1 Report
The authors have responded to all my comments and make recommended corrections to all except 1. When I asked to add about the effectiveness of nanoemulsions, I meant their effectiveness in behavioral tests on rats. There is information in the text about improved pharmacokinetics, but the improved behavior of rats after administration of nanoemulsions and reduced depressive-like behavior is still only mentioned 1 time. Only lines 350 to 357 describe behavioral tests on rats. Please add more information how nanoemulsions improve the behavior of rats and reduce their depressive-like state.
Author Response
We thank the reviewer for their constructive criticism and expert opinion. The manuscript has been altered and improved according to the reviewer’s suggestions (changes marked by coloring in pink). The therapeutic effectiveness of nanoemulsions in animal models, aka pharmacodynamic behavioral tests, was in fact only done in two studies. We have now completed the existing information, and it can be found from lines 324 to 333, and 520 to 526. Nevertheless, these types of tests are in fact lacking, but extremely important, and hence we have added a paragraph from lines 676 to 681 stating that this is a limitation.
Reviewer 2 Report
The manuscript was improved and can be accepted for publication
Author Response
We thank the reviewer for their constructive criticism and expert opinion, which has helped better our manuscript, and thank their approval for its publication in Pharmaceutics.